# Phenotypic Heterogeneity of Cancer Associated Fibroblasts in Cervical Cancer Progression: FAP as a Central Activation Marker

**DOI:** 10.3390/cells13070560

**Published:** 2024-03-22

**Authors:** Lesly Jazmin Bueno-Urquiza, Marisol Godínez-Rubí, Julio César Villegas-Pineda, Alejandra Natali Vega-Magaña, Luis Felipe Jave-Suárez, Ana Graciela Puebla-Mora, Gloria Estefanía Aguirre-Sandoval, María Guadalupe Martínez-Silva, Adrián Ramírez-de-Arellano, Ana Laura Pereira-Suárez

**Affiliations:** 1Instituto de Investigación en Ciencias Biomédicas, Centro Universitario de Ciencias de la Salud, Universidad de Guadalajara, Guadalajara 44340, Mexico; lesly.buenou@gmail.com (L.J.B.-U.); alejandra.vega@academicos.udg.mx (A.N.V.-M.); adrian.ramirez@academicos.udg.mx (A.R.-d.-A.); 2Departamento de Microbiología y Patología, Centro Universitario de Ciencias de la Salud, Universidad de Guadalajara, Guadalajara 44340, Mexico; juliana.godinez@academicos.udg.mx (M.G.-R.); julio.villegas@academicos.udg.mx (J.C.V.-P.); graciela.puebla@academicos.udg.mx (A.G.P.-M.); fani4419@gmail.com (G.E.A.-S.); 3División de Inmunología, Centro de Investigación Biomédica de Occidente, Instituto Mexicano del Seguro Social, Guadalajara 44340, Mexico; lfjave@gmail.com; 4Departamento de Anatomía Patológica, Centro Médico Nacional de Occidente, Instituto Mexicano del Seguro Social (IMSS), Guadalajara 44340, Mexico; lupita.doctora694@gmail.com

**Keywords:** cancer-associated fibroblasts, fibroblast activation protein, cervical cancer, mesenquimal stem cells, heterogeneous phenotypes

## Abstract

Cervical cancer (CC) is the fourth leading cancer among women and is one of the principal gynecological malignancies. In the tumor microenvironment, cancer-associated fibroblasts (CAFs) play a crucial role during malignant progression, exhibiting a variety of heterogeneous phenotypes. CAFs express phenotypic markers like fibroblast activation protein (FAP), vimentin, S100A4, α-smooth muscle actin (αSMA), and functional markers such as MMP9. This study aimed to evaluate the protein expression of vimentin, S100A4, αSMA, FAP, and MMP9 in mesenchymal stem cells (MSC)-CAF cells, as well as in cervical cancer samples. MSC cells were stimulated with HeLa and SiHa tumor cell supernatants, followed by protein evaluation and cytokine profile to confirm differentiation towards a CAF phenotype. In addition, automated immunohistochemistry (IHQa) was performed to evaluate the expression of these proteins in CC samples at different stages. Our findings revealed a high expression of FAP in stimulated MSC cells, accompanied by the secretion of pro/anti-inflammatory cytokines. In the other hand, CC samples were observed to have high expression of FAP, vimentin, αSMA, and MMP9. Most importantly, there was a high expression of their activation proteins αSMA and FAP during the different stages. In the early stages, a myofibroblast-like phenotype (CAFs αSMA+ FAP+), and in the late stages a protumoral phenotype (CAF αSMA− FAP+). In summary, FAP has a crucial role in the activation of CAFs during cervical cancer progression.

## 1. Introduction

As of 2020, cervical cancer (CC) stands as the fourth most prevalent cancer among women worldwide, with approximately 604,000 new cases and 342,000 deaths, marking it as a critical area of gynecological research and intervention [1]. This malignancy is classified into four primary stages by the International Federation of Gynecology and Obstetrics (FIGO) and the TNM classification system, which assesses tumors’ sizes and extents [2]. Unlike other gynecological neoplasms, in cervical cancer persistent infection by human papillomavirus (HPV) is crucial and is responsible for 97% of cases [3].

The tumor microenvironment (TME) is composed of a variety of cellular and extracellular factors that shapes the tumor stroma. Which is composed of cytokines, growth factors, extracellular matrix (ECM), and predominantly cell populations such as cancer-associated fibroblasts (CAFs) [4,5]. CAFs are categorized as the “tumor masters/orchestrators” due to their ability to secrete cytokines and growth factors involved in the recruitment or polarization of other cell populations during tumor progression [6].

Recent research has highlighted the heterogeneity of cancer-associated fibroblasts (CAFs), as demonstrated by the expression of various markers indicative of their diverse origins [7]. These cells can originate from mesenchymal stem cells (MSC), which possess the capacity for self-renewal and multipotentiality, enabling them to differentiate into various cell types, including adipocytes, chondrocytes, osteoclasts, and fibroblasts [8,9]. MSCs can differentiate into CAFs upon exposure to tumor cells [10]. Additionally, the expression of several proteins has been observed in CAFs throughout tumor progression. Among these markers are fibroblast proteins like vimentin and S100A4, whose expression is elevated in cancer. However, one of the most notable activation markers is α-smooth muscle actin (αSMA), which imparts contractile functions and elevated cellular motility to CAFs, giving them a myofibroblast-like phenotype [11].

Once activated, CAFs express significant levels of fibroblast activation protein (FAP), a type II transmembrane serine protease with specific endopeptidase activity. This enzymatic activity contributes to higher protumoral effects [12]. Consequently, CAF populations with elevated FAP expression are linked to reduced survival rates and more aggressive cancer phenotypes [13]. Furthermore, CAFs regulate tumor invasion, with metalloproteinases playing a key role in this mechanism. MMP9, a zinc metalloproteinase, is associated with invasion in various tumors due to its ability to degrade the extracellular matrix in the TME [14].

Despite the diversity of CAF phenotypes observed in various types of cancer, there are still no studies investigating the presence of heterogeneous CAF phenotypes during cervical cancer progression. For these reasons, this study evaluated the effect of supernatants from cervical cancer cell lines (HeLa and SiHa) in the expression of vimentin, S100A4, αSMA, FAP, and MMP9 markers in an in vitro MSC-CAF model. In addition, we investigated the protein levels of vimentin, S100A4, αSMA, FAP, MMP9, and Ki67 in different stages of cervical cancer.

## 2. Materials and Methods

### 2.1. Cell Culture Conditions

SiHa and HeLa are cell lines derived from cervical cancer obtained from the ATCC^®^ (American Type Culture Collection, Manassas, VA, USA). These cell lines were culture in DMEM GlutaMAX (Cat. No. 10566016, Gibco™, Waltham, MA, USA) supplemented with 10% Fetal Bovine Serum (FBS, Cat. No. 26140-079, Gibco™, Waltham, MA, USA) and penicillin (10,000 U/mL), streptomycin (10,000 μg/mL) and amphotericin B (25 µg/mL) (No. Cat. 15240062, Gibco) in a humidified atmosphere with 5% CO_2_ at 37 °C. 

Mesenchymal stem cell line (MSCs) (Cat. No. PCS-500-012, ATCC^®^, Manassas, VA, USA) culture in DMEM/F-12 GlutaMAX (Cat. No. 10565018, Gibco™, Waltham, MA, USA) supplemented with a special growth kit for this cell line (Cat. No. PCS500041, ATCC^®^, Manassas, VA, USA) and antibiotic in a humidified atmosphere with 5% CO_2_ at 37 °C.

### 2.2. Supernatant Collection of Tumor Cells and Stimulation of MSCs 

HeLa and SiHa cells were cultured in DMEM GlutaMAX until they obtained 90% confluence. Once this confluence was reached, the general medium was replaced with a serum-free medium. After 48 h, the supernatant was collected and used as conditioned medium, which was then stored at −80 °C.

Once the MSCs reached 90% confluence, they were cultured with conditioned medium from cervical cancer cells (HeLa and SiHa) for 48 h. The supernatants from these MSC cultures were collected for subsequent experiments.

As a negative control, MSCs were cultured with medium without serum, while as a positive control, MSCs were stimulated with the cytokine transforming growth factor beta (TGF-β) at a concentration of 20 ng/mL (Cat. No. 240-B-002, R&D systems, Minneapolis, MN, USA).

### 2.3. Characterization of MSC-CAFs

The differentiation of MSC cells towards a CAF phenotype was performed by immunofluorescence. Initially, 5000 cells were plated, washed with PBS, and fixed with 4% paraformaldehyde for 10 min at room temperature. Cells were permeabilized with 0.2% Tween 20 (Cat. No. P1379-500ML) for 10 min. Subsequently, a blocking step was carried out using PBS containing 10% FBS (FBS, Cat. No. 26140-079, Gibco™, Waltham, MA, USA) and 1% BSA (Cat. No. A2153-50G, Sigma Aldrich, St. Louis, MI, USA) for 1 h at 37 °C. The primary antibodies used were: Vimentin (Cat. No. RV202, Abcam, Cambridge, MA, USA; 1:500 dilution), αSMA (Cat. No. ab7817, Abcam, Cambridge, MA, USA; 1:500 dilution), FAP (Cat. No. ab28244, Abcam, Cambridge, MA, USA; 1:200 dilution), S100A4 (Cat. No. ab124805, Abcam, Cambridge, MA, USA; 1:100 dilution, 1:500) and MMP9 (Cat. No. ab76003, Abcam, Cambridge, MA, USA) antibodies. The incubation with the primary antibody was performed overnight at 4 °C, followed by incubation with secondary antibodies (anti-rabbit IgG/Alexa Fluor™ 488 (Goat IgG; Cat. No A-11008, Invitrogen™, Waltham, MA, USA)) and anti-mouse IgG/FITC (Goat IgG; Cat. No. ab6785, Abcam, Cambridge, MA, USA), both at a 1:1000 dilution for 2 h. The nuclei were stained with DAPI (Cat. No. D1306, Invitrogen™, Waltham, MA, USA; 1:15,000 dilution) for 5 min in the dark. As a positive control, MSC cells were stimulated with TGFβ (20 ng/mL) for 48 h before incubation with primary and secondary antibodies. 

The slides were observed in an Axio Imager 2 fluorescence microscope (Carl Zeiss, Göttingen, Germany), using filters with the following excitation ranges: Alexa Fluor excitation: 495 nm, emission: 519 nm; DAPI excitation: 351 nm; emission: 461 nm. We analyzed the images using ImageJ software (version 1.53k, National Institutes of Health, Bethesda, MD, USA, Public Domain). ImageJ software version number #1.54i. For each sample, we examined five fields, with 30 cells per field, all in TIFF format. Additionally, we utilized the software to generate a binary mask and a region of interest (ROI) manager to precisely select the area of each cell. This enabled us to measure the intensity or integrated density (“IntDen”) of each cell per replica, which is essential for conducting statistical, we used one-way ANOVA. 

### 2.4. Cytokine Profile of MSCs-CAFs

To determine the cytokine profile of MSC-CAF, the supernatant was used. We performed the analysis using the Bio-Plex^®^ Multiplex Immunoassays (BIO-RAD) system. This system consists in fluorescently dyed microspheres, each with a unique color code or spectral address, enable the discrimination of individual tests within a multiplex suspension. This allows for the simultaneous detection of up to 500 different types of molecules in a single well of a 96-well microplate. Specifically, we used the Bio-Plex Pro Human Cytokine 17 (Cat. No. 10023381; G-CSF, GM-CSF, IFN-, IL-1β, IL-2, IL-4, IL-5, IL-6, IL-7, IL-8, IL-10, IL-12, IL-13, IL-17A, MCP-1, MIP-1β, and TNF-α) and Bio-Plex Pro™ TGF-β 3-plex Assay (Cat. No. 171W4001M; TGF-β1, TGF-β2, and TGF-β3) kits from (Bio-Rad, Hercules, CA, USA) following the manufacturer’s instructions. The cytokine concentration was determined using the MAGPIX^®^ flow system (Bio-Rad, Hercules, CA, USA) and expressed in pg/mL. For the analysis, we examined only one replicate of MSC supernatants. However, these supernatants were randomly selected from all those collected for the study.

### 2.5. Cervical Cancer Patients

Paraffin-embedded cervical cancer and healthy cervical tissue samples, archived by the Pathological Anatomy service from 2017 to 2021, were included in the study. We obtained a total of 31 samples, these cases included cancer in situ (9), early stages I–II (11), and late stages III–IV (11). These cases were staged according to the FIGO staging system, which were examined and selected based on tissue integrity. 

### 2.6. Automated Immunohistochemistry

Tissues were previously fixed with 10% formaldehyde and embedded in paraffin. Sections were made and mounted on electrocharged slides (Cat. No. 3800200 White 1′′ × 3′′ × 1.0 mm; Leica Biosystems, Nußloch, Germany), then deparaffinized and dehydrated. The samples were processed automatically in the BOND MAX equipment (Leica Biosystems). The BOND Polymer Refine Detection Kit (Cat. No. DS9800) was used for immunodetection. The primary antibodies used were: vimentin (Cat. No. RV202, Abcam, Cambridge, MA, USA; 1:1000 dilution), αSMA (Cat. No. ab7817, Abcam, Cambridge, MA, USA; 1:15,000 dilution), FAP (Cat. No. ab28244, Abcam, Cambridge, MA, USA; 1:100 dilution), S100A4 (Cat. No. ab124805, Abcam, Cambridge, MA, USA; dilution 1:1000) and MMP9 (Cat. No. ab76003, Abcam, Cambridge, MA, USA, dilution 1:1000) Ki67 (cat no. ab15580; Abcam, dilution 1:100). For the negative control, we omitted the primary antibody to exclude the possibility of non-specific binding. Finally, they were assembled and sealed with synthetic resin (Cat No. 7987, HYCEL). 

Analysis of the immunohistochemistry (IHQ) assays performed visually by two certified pathologists. With 10× magnification, the areas of maximum intensity were identified, and subsequently, 5 fields of each sample were analyzed at 40×. Later to corroborate the pathologists’ results, the slides were scanned and digitized with the Aperio LV1 real-time pathology system (Leica Biosystems) and subsequently analyzed using the QuPath 0.5.0 program. Five fields of 200 × 200 microns were taken into account in a maximum area of protein positivity, and both are expressed as percentage of positive cells. We conducted a normality analysis for the sample data, and based on the results, we chose to use one-way ANOVA. Each bar in the graphs represents the percentage of positive cells, with data variability indicated by the standard deviation. 

### 2.7. Statistical Analysis

The analysis of vimentin, S100A4, αSMA, FAP, MMP9, and Ki67 expression was performed using GraphPad Prism 8.0.2 and RStudio version 4.3.2 statistical software. One-way ANOVA was independently performed for each protein to assess differences in expression levels across different clinical stages. Post hoc Tukey tests were used to compare the mean percentage of positive cells in healthy cervix samples with the mean of each clinical stage. Furthermore, a Spearman correlation analysis was carried out in RStudio to explore associations between the expression levels of vimentin, S100A4, αSMA, FAP, Ki67, and MMP9. A significance level of *p* < 0.05 was considered statistically significant for all analyses.

## 3. Results

### 3.1. Expression of CAF-Associated Proteins in MSC Cells Stimulated with Cervical Cancer Cell Supernatants

To determine whether cervical cancer cells conditioned media (SiHa and HeLa) can promote differentiation towards an MSC-CAF phenotype we evaluated vimentin, S100A4, αSMA, FAP, and MMP9 expression by immunofluorescence (Figure 1 and Figure 2).

Cervical cancer cells conditioned media from HeLa and SiHa cells do not induce vimentin expression significantly; however, the integrated density of expression is very strong in the cytoplasmic region of stimulated MSCs. When analyzing the expression of S100A4, another mesenchymal marker, we observed that only the supernatants of HeLa cells promoted high expression of this protein (*p* = 0.0100) but with a lower fluorescence intensity than vimentin (Figure 1b).

When investigating whether cervical cancer cells conditioned media lead to a heterogeneous MSC activation phenotype, we found a high fluorescence intensity of αSMA and FAP indicating a differentiation towards an MSC-CAF phenotype.

Cervical cancer cells conditioned media did not induce changes in the expression level of αSMA, a hallmark of myofibroblasts, compared to unstimulated MSCs, despite the high integrated fluorescence density presented (Figure 2b). However, these supernatants promoted a significantly higher expression in FAP, a CAF specific marker, (*p* ≤ 0.0001) in MSC cells (Figure 2b). Additionally, we used supernatants from the non-tumorigenic keratinocyte cell line (HaCaT) to stimulate MSCs and we evaluated the expression of the proteins vimentin and αSMA. As expected, we observed a strong expression of vimentin as it is a mesenchymal marker, however the expression of the myofibroblast activation marker αSMA was very weak. 

When examining a functional marker of CAF such as MMP9, we observed a low expression in the cytoplasmic region without significant differences with stimulated cells (Figure 2c). 

### 3.2. Cytokine Production from MSCs Stimulated with Cervical Cancer Cell Supernatants

Our prior findings suggest that cervical cancer cells conditioned media (HeLa and SiHa) influence differentiation towards a CAF phenotype. To provide additional evidence of this differentiation under the influence cervical cancer cells conditioned media, we assessed cytokine production using a multiplex assay.

Cytokines in the tumor microenvironment play a crucial role in cell communication. In our experiment, we observed that cytokines predominantly produced by stimulated MSCs are associated with immune cell recruitment and polarization processes.

Despite both cervical cancer cells conditioned media, they elicit different effects on MSC cytokine production. HeLa supernatants induce higher TNFα expression (7.64 pg/mL) compared to SiHa supernatants. Conversely, G-CSF production (1.14 pg/mL) exhibits an opposite effect (Figure 3a). No changes were observed with either supernatant for other cytokines associated with cell recruitment, such as GM-CSF, IL-8, CCL2, and CCL4.

Concerning cytokines involved in cell polarization, IL-6 (0.36 pg/mL), IFN-γ (0.68 pg/mL), and IL-10 (0.68 pg/mL) production increased in MSC cells only under the stimulus of HeLa supernatants. However, IL-7 (10.17 pg/mL) increased in the presence of SiHa supernatant stimulation, while TGF-β1 (9031.23 pg/mL) decreased with both supernatants (Figure 3b). Conversely, the production of IL-1b, IL-2, IL-5, IL-12, IL-13, and IL-17a did not exhibit changes with any of the stimuli.

### 3.3. Expression of CAF-Associated Proteins (Vimentin, S100A4) in Tissues of Cervical Cancer

To evaluate the expression of proteins associated with a CAF phenotype previously determined in vitro, we performed automated immunohistochemistry (IHQa) on paraffin-embedded tissue samples from different stages of cervical cancer.

Out of the 31 cases of cervical cancer analyzed, 9 of them (29%) corresponded to carcinoma in situ, 11 (35.48%) to early stages (stages I, II), and 11 (35.48%) to late stages (stages III, IV); there were an additional 10 control samples. Regarding the histological type, 15 of the tumors (68.18%) corresponded to squamous cell carcinoma, 5 (22.72%) to adenocarcinoma, 1 (4.54%) to adenosquamous carcinoma, and 1 (4%) to neuroendocrine carcinoma. The age range of patients and controls is 40 to 60 years (Table 1).

To demonstrate the presence of cells of mesenchymal origin, we evaluated the expression of vimentin. Our results showed that vimentin expression is significantly increased in early-stage (*p* = 0.02) and late-stage (*p* = 0.02) cervical cancer compared to samples with in situ cancer and those used as controls. However, there were no significant differences between stages (Figure 4a).

On the other hand, S100A4 expression showed no significant difference in the percentage of positive cells in cases with cancer vs. control (Figure 4b).

### 3.4. Expression of CAF-Associated Proteins (αSMA, FAP, Ki67, MMP9) in Tissues of Cervical Cancer

To assess the activation of CAFs in tissue samples, we performed immunostaining for αSMA and FAP.

The expression of αSMA is significantly increased in samples of in situ cancer (*p* = 0.009) and early-stages (I, II) (*p* = 0.0026) compared to samples without cancer. In late stages (III, IV), we observed an increasing trend in the expression of this protein; however, we did not find any significance (Figure 5a).

Conversely, FAP expression was significantly increased in all cancer stages compared to controls (*p* < 0.0001). Notably, FAP expression was observed in both CAF and tumor cells, in contrast to other markers that are predominantly expressed in CAFs (Figure 5b).

In addition to assessing the activation of CAFs, we analyzed the expression of two proteins associated with proliferation (Ki67) and invasion (MMP9).

Regarding Ki67, we observed higher expression only in late stages (III, IV) (*p* = 0.0006) compared to other groups (Figure 5c). MMP9 expression was significantly higher in both early stages (*p* = 0.0053) and late stages (*p* = 0.0001) and increased progressively as the lesion (Figure 5d).

### 3.5. Association of CAFs related-proteins with Proliferation and Invasion in Tissues of Cervical Cancer

To assess the relationship between proliferation (Ki67) and invasion (MMP9) with the expression of CAFs markers, including vimentin, S100A4, αSMA, and FAP, we conducted a Spearman correlation analysis (Figure 6).

We observed a negative association between Ki67 and MMP9 expression in carcer in situ (Figure 6a). However, in the early and late stages of cervical cancer, we found no significant correlation between the expression of CAF markers and Ki67 or MMP9 (Figure 6b,c). In advanced stages of cervical cancer, we found that there is an overexpression of FAP and MMP9 with a positive trend (Figure 6c).

## 4. Discussion

The cervical tumor microenvironment comprises a diversity of cell populations that interact with each other to either positively or negatively impact the tumor [4]. Among these populations, CAFs play a crucial role in the progression of various types of tumors, including cervical, breast, colon, and pancreatic cancer. They regulate diverse mechanisms such as proliferation, migration, apoptosis, invasion, metastasis, and resistance to therapies [15,16,17,18,19].

This important cell population is heterogeneous due to the diversity of its origins and therefore can present diverse phenotypes. Similarly, studies have reported that mesenchymal stem cells obtained from bone marrow can differentiate into a CAF phenotype when cultured with cancer cell lines [10]. Our results demonstrate that supernatants from cervical cancer cell lines (HeLa and SiHa) can promote the differentiation of MSCs into a CAF phenotype by inducing the overexpression of proteins such as αSMA and FAP.

These findings are consistent with those reported in other tumor types, where they observed that MSCs stimulated with conditioned medium from colon, breast, and pancreatic cell lines overexpressed αSMA, S100A4, and vimentin [10,20]. Additionally, a distinct subset of CAFs isolated from colon cancer demonstrated increased expression of the FAP protein when co-cultured with colon cancer cell lines [21]. Conversely, another study demonstrated that CAFs isolated from highly aggressive pancreatic tumors exhibited elevated levels of this protein [22].

Cytokines produced in the TME by immune and tumor cells modulate fundamental processes that regulate the development of CC [23]. HeLa and SiHa cell lines secrete large amounts of IL-6 and TNFα, which induce differentiation to a protumor M2 phenotype of macrophages [24].

Within the TME, CAFs are among the primary cytokine-producing cells that regulate cell migration, recruitment, and polarization responses. The MSC-CAFs stimulated with supernatants of HeLa and SiHa cells exhibit secretion of various cytokines, predominantly IL-6, TNFα, G-CSF, IFN-γ, IL-10, and IL-7. These cytokines play crucial roles in the aforementioned processes. Some of these cytokines show increased levels in patients with cervical cancer compared to those with premalignant lesions and controls, indicating alterations in the cervical immune microenvironment [23,25].

Additionally, it has been shown that CAFs from colon cancer promote IL8/CXCR2-mediated monocyte adhesion and attraction through upregulation of VCAM-1 expression and IL-6 production, and subsequently promote polarization to an M2 phenotype by expression of CD163 and CD206 [26].

A similar study reported the effect of CAF obtained from oral squamous cell cancer cells (OSCC) on the polarization of macrophages to a protumoral M2 profile expressing high amounts of IL-10, which in turn suppresses T-cell proliferation [27].

On the other hand, CAFs derived from neuroblastomas have shown that IL-6 activates various signaling pathways, including STAT3, ERK1/2, and STAT1, in tumor cells [28]. This emphasizes the immunoregulatory role of cytokines produced by various cells in the TME and indicates the importance of understanding the mechanisms influencing tumor development and progression.

In the context of cervical cancer, one crucial factor contributing to tumor progression is HPV infection. In relation to vimentin expression, it has been reported that both HPV 16 and HPV 18 can induce epithelial-mesenchymal transition (EMT), which causes a change in the morphology of keratinocytes, acquiring a fibroblastoid phenotype [29,30]. Understanding how cells within the TME communicate in this environment is essential for elucidating the pathophysiology of this neoplasm [2].

Given the heterogeneity of CAFs and the lack of evidence for the presence of different phenotypes in this neoplasm, in our study, we investigated the presence of CAF phenotypes during the progression of CC.

Specifically, we observed that CAFs differentially express some activation proteins, such as αSMA and FAP during CC progression. In early stages, we observed CAFs with a myofibroblast-like phenotype (CAFs αSMA+ FAP+) and in late stages CAFs with a protumoral phenotype (CAF αSMA− FAP+).

These phenotypes are similar to those found in pancreatic cancer, where two subpopulations of CAFs were identified based on their tumor location (classified as either near or far from the tumor) and functionality, as determined by their cytokine expression profiles. The myofibroblast-like CAFs αSMA+ (myCAFs) were predominantly located proximal to the tumor, while the inflammatory CAFs FAP+ (iCAFs) were positioned distally from the tumor. This last subset exhibited increased secretion of IL-6 and CCL2, which play crucial roles in regulating immune cell recruitment [31,32].

Another example of characterization of CAF phenotypes was performed on different molecular subtypes of breast cancer, where four subpopulations with different properties and activation levels were observed. These subpopulations were classified as follows: CAFS1 (expressing αSMA, FAP, PDGFRβ, CD29, and FSP1/S100A4), CAFS2 (expressing CD29 and FSP17S100A4), CAFS3 (expressing αSMA, PDGFRβ, CD29, and FSP1/S100A4), and CAFS4 (expressing αSMA, PDGFRβ, CD29, and FSP1/S100A4). CAFS1 and CAFS4 were predominantly found in triple-negative and HER2+ breast cancers, while CAFS2 was associated with the LumA subtype, and CAFS3 was present across all three subtypes [33].

In another work, they also found four CAF populations in breast cancer, but their markers were differentially expressed, CAFS1 and CAFS4 (FAP, CD29, αSMA, PDGFRβ), CAFS2 (αSMA, PDGFRβ), CAFS3 (CD29, PDGFRβ). Notably, CAF phenotypes positive for FAP (CAFS1 and CAFS4) were associated with processes related to tumor invasion and metastasis [34]. Similarly, our findings revealed increased FAP protein expression during cancer progression, suggesting its involvement in these protumoral processes.

The FAP protein is a serine protease with endopeptidase activity and has been associated with poor prognosis in various tumors, such as breast, colorectal, oral, lung, and ovarian cancer [13,35,36,37]. Studies have revealed that genetic suppression or pharmacological inhibition of this protein has a significant impact on the decrease in lung cancer cell proliferation in murine models. This tumor growth is affected by changes in the composition or organization of the extracellular matrix (ECM) mediated by dysregulation of integrin signaling [38]. Furthermore, it has been observed that these events of proliferation, migration, invasion, and metastasis are modulated by FAP through the PTEN/PI3K/Akt and Ras-ERK pathways in OSCC cells [37,39,40]. 

Furthermore, studies have demonstrated that FAP can be transcriptionally upregulated by the canonical TGF-β pathway using an in vitro model of melanoma [41].

This leads us to understand that CAFs subpopulation with increase FAP expression promotes higher tumor aggressiveness and immunosuppression [33,34]. In cervical lesions, it has been reported that FAP increases significantly in advanced premalignant lesions and cancer. Furthermore, it has been observed that Wnt2B released from exosomes of tumor cells positively regulates α-SMA and FAP expression in vitro and in vivo and these acquire a greater capacity for proliferation and migration [16].

As expected, the proliferation associated protein Ki67 was found to be overexpressed in advanced stages (III, IV), consistent with observations in other tumors [42,43].

Similarly, our analysis revealed increased expression of MMP9 as the lesion progressed, which associated with FAP expression. In other tumors as well, CAFs have been found to express large amounts of MMP9 [44,45]. In breast cancer, tumor cells were reported to induce MMP9 expression in fibroblasts through Smad, Ras, MAPK, and PI3-kinase pathways in models in vitro of co-culture of tumor cells and fibroblasts [46].

## 5. Conclusions

In conclusion, our findings indicate that the FAP+ CAF population is linked to increased tumor progression in cervical cancer, underscoring the pivotal role of the FAP protein in the processes of proliferation and invasion. However, a limitation at this stage of the study was that we were not able to obtain fresh biopsies from patients with cervical cancer to isolate CAFs and thus be able to perform functional studies. Currently, work is underway to evaluate the crosstalk of CAF and tumor cells. Therefore, future research should investigate its utility as a prognostic marker with significant clinical implications for cervical cancer.

## Figures and Tables

**Figure 1 cells-13-00560-f001:**
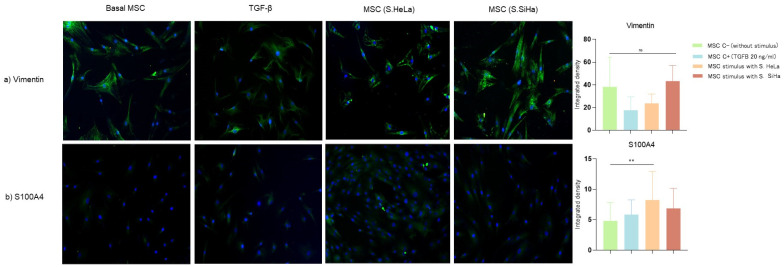
Expression of vimentin, S100A4, in MSC cells without stimulus and stimulated with supernatant from SiHa and HeLa cells. Vimentin and S100A4 expression were evaluated through immunofluorescence using an FITC-conjugated secondary antibody (green) and nuclear staining with DAPI (blue). Merged images are shown at 10× magnification. (**a**) Expression of vimentin in MSC cells; (**b**) Expression of S100A4 in MSC cells. We used ImageJ for the quantification. Differences were found in S100A4 in the groups: MSC C− vs. MSC stimulated with S. HeLa. Three independent experiments were performed. Statistical analysis was performed using one-way ANOVA. Data are presented as mean ± SD. ** *p* < 0.05, ns: non-significant followed by a Tukey test. MSC C−: MSC unstimulated (basal), MSC C+: stimulated with TGFβ (20 ng/mL), MSC stimulated with S. HeLa: MSC stimulated with supernatant from HeLa, MSC stimulated with S. SiHa: MSC stimulated with supernatant from SiHa.

**Figure 2 cells-13-00560-f002:**
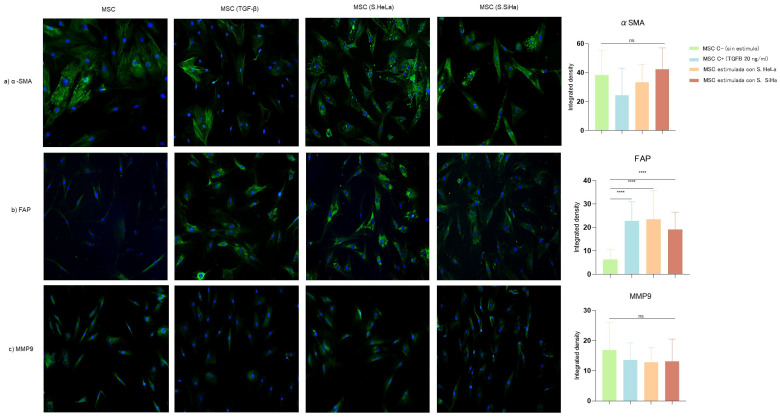
Expression of αSMA, FAP, and MMP9 in MSC cells unstimulated and stimulated with supernatant from SiHa and HeLa cells. The αSMA, FAP, and MMP9 expression were evaluated through immunofluorescence using an FITC-conjugated, Alexa488-conjugated secondary antibody (green) and nuclear staining with DAPI (blue). Merged images are shown at 10× magnification. (**a**) Expression of αSMA in MSC cells; (**b**) Expression of FAP in MSC cells; (**c**) Expression of MMP9. Differences were found in FAP in the groups: MSC C− vs. MSC + (TGFβ), MSC stimulated with S. HeLa, MSC stimulated with S. SiHa. Three independent experiments were performed. Statistical analysis was performed using one-way ANOVA. Data are presented as mean ± SD, **** *p* < 0.0001, ns: non-significant followed by a Tukey test. MSC C−: MSC unstimulated (basal), MSC C+: stimulated with TGFβ (20 ng/mL), MSC stimulated with S. HeLa: MSC stimulated with supernatant from HeLa, MSC stimulated with S. SiHa: MSC stimulated with supernatant from SiHa.

**Figure 3 cells-13-00560-f003:**
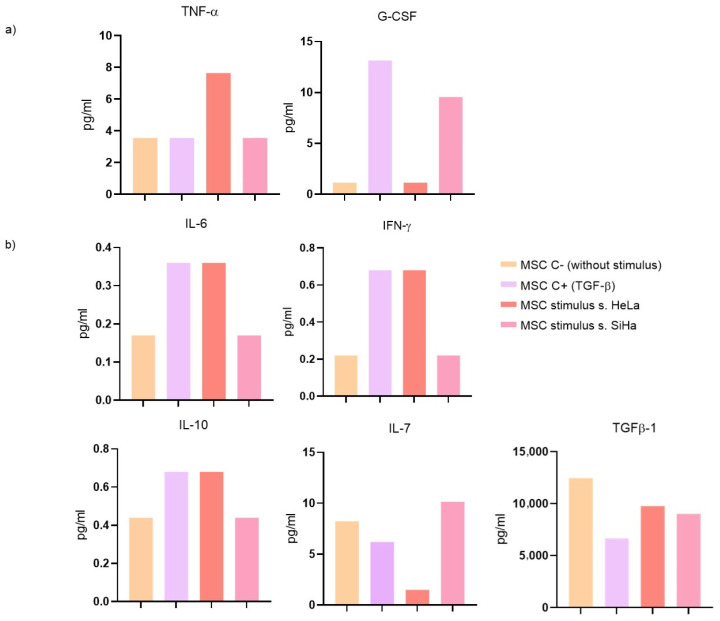
Cytokine production in MSC cells stimulated with HeLa and SiHa supernatants. (**a**) recruitment cytokines (TNF-α, G-CSF); (**b**) polarization cytokines IL-6, IFN-γ, IL-10, IL-7, and TGF-β1 using a multiplex immunoassay. The results of an independent assay are presented in pg/mL for each cytokine: MSC: MSC unstimulated (basal), MSC +: stimulated with TGFβ (20 ng/mL), MSC stimulated with S. HeLa: MSC stimulated with supernatant from HeLa, MSC stimulated with S. SiHa: MSC stimulated with supernatant from SiHa.

**Figure 4 cells-13-00560-f004:**
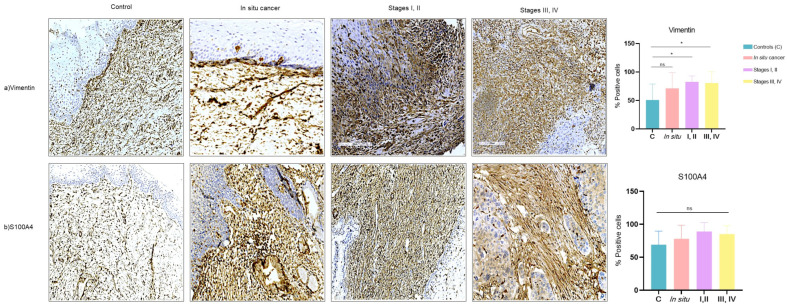
Representatives images of expression of vimentin and S100A4 in cervical tissues from controls (cervix without CC), in situ cancer, stages I, II (CC I, II), Stages III, IV (CC III, IV). (**a**) vimentin, (**b**) S100A4, were detected by automated immunohistochemistry (IHQa); brown staining indicates positive expression. Statistical analysis was performed in % positive cells using one-way ANOVA in five fields. Data are presented as mean ± SD. * *p* < 0.05, ns: non-significant followed by a Tukey test. Each of the panels are representative photomicrographs of five fields (200 microns).

**Figure 5 cells-13-00560-f005:**
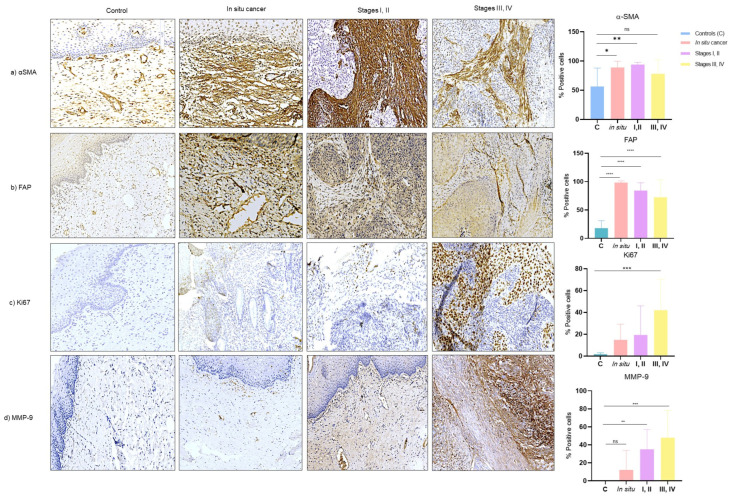
Representatives images of expression of αSMA, FAP, ki-67, and MMP9 in cervical tissues from controls (cervix without CC), in situ cancer, stages I, II (CC I, II), stages III, IV (CC III, IV). (**a**) αSMA, (**b**) FAP, (**c**) KI67, and (**d**) MMP9 were detected by automated immunohistochemistry (IHQa); brown staining indicates positive expression. Statistical analysis was performed in % positive cells using one-way ANOVA in five fields. Data are presented as mean ± SD. * *p* < 0.05, ** *p* < 0.05, **** *p* < 0.0001, *** *p* < 0.0001, ns: non-significant followed by a Tukey test. Each of the panels are representative photomicrographs of five fields (200 microns).

**Figure 6 cells-13-00560-f006:**
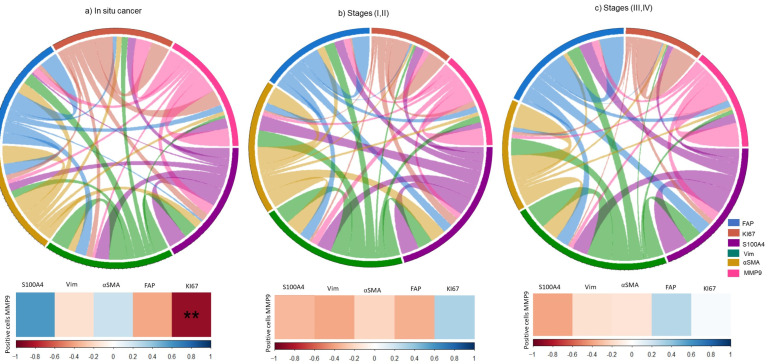
Correlogram between vimentin, S100, αSMA, FAP, MMP9, and Ki67 in samples with CC. The upper section displays a chord diagram, where each protein is represented by a different color, visually depicting their relationships in each stage. ***p* < 0.05; The thickness of the chord indicates the strength of the association between proteins. The lower section shows a correlation matrix, indicating the relationships between the proteins and the percentage of MMP9-positive cells. Positive correlations are highlighted in blue, while negative correlations are denoted in red. The analysis was performed using Spearman’s correlation in RStudio.

**Table 1 cells-13-00560-t001:** Clinicopathologic characteristics of the patients and control groups.

Control Groups		Patients with Confirmed Diagnosis			
*n* = 10 42–72 years		*n* = 31 25–85 years			
Histopathological diagnosis	%	Clinical stage	%	Histological types	%
Cystic chronic cervicitis	7(70%)	In situ cancer	9 (29%)	Squamous cell carcinoma	15 (68.18%)
Active chronic cervicitis with ulceration	1 (10%)	Early stages (I, II)	11 (35.48%)	Adenocarcinoma	5 (22.72%)
Chronic acute cervicitis with ulceration	1 (10%)			Mixed carcinoma (adenosquamous)	1 (4.54%)
Chronic mild cervicitis	1 (10%)	Late stages (III, IV)	11 (35.48%)	Neuroendocrine carcinoma	1 (4.54%)

Data are shown as mean and range and number (%).

## Data Availability

The datasets used and/or analyzed during the current study are available from the corresponding author upon reasonable request.

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
