# Peer review of "Phenotypic Heterogeneity of Cancer Associated Fibroblasts in Cervical Cancer Progression: FAP as a Central Activation Marker"

_cells, 2024, doi:10.3390/cells13070560_

Round 1

Reviewer 1 Report

Comments and Suggestions for Authors

The results from this study are interesting. Using the cells and cervical cancer samples, authors showed that high expression of FAP in stimulated MSC cells. In the Cervical cancer samples they observed high expression of FAP, aSMA, vimention and MMP9. At early stages of cervical cancer, myofibroblast-like phenotype (CAFs aSMA+FAP+) and at late stages protumoral phenotype (CAFs aSMA-FAP+) was observed. They concluded that the CAFs FAP+ population is associated with increase tumor progression in cervical cancer indicating the role of FAP protein in proliferation and invasion processes. The future studies are to explore the potential role of FAP as prognostic marker with clinical relevance for cervical cancer.

11.  In the manuscript some places it is mentioned the effect of supernatants from cervical cancer cell lines and in other places it is mentioned as cervical cancer cells conditioned media. It is appropriate to use cervical cancer cells conditioned media to follow the consistency in all the places.

22.  Are the correct isotype controls were used while performing the IHC? In the methods section this information in not provided.

F3. Figure 1 and 2 the labels for integrated density are not showing, need to be fixed.

44.  Results are discussed well in the discussion section. Including one aspect on HPV will be useful. Is there a literature on the effect of HPV on Vimentin, aSMA and FAP and MMP9? This need to be discussed in the discussion section. Since the studies are focused on TME and used the cervical cancer cells conditioned media, the effect of HPV on the markers needs to be discussed.

Reviewer 2 Report

Comments and Suggestions for Authors

Comments on the Quality of English Language

Minor editing of English language required. As a non-native English speaker, I have suggested some reformulations to improve the language in the manuscript. However, I recommend considering a review by a native English speaker to further enhance sentence construction, readability, coherence, and accuracy, ensuring a polished final presentation.
